# Electronic based reported anthropometry—A useful tool for interim monitoring of obesity prevalence in developing states

**Pamela S. Gaskin**[1]*, **Peter Chami**[2,3], **Tamara Nancoo**[1], **Patricia Warner**[4], **Patrick Barrett**[3], **Yvette Mayers**[5]

**1** Faculty of Medical Sciences, University of the West Indies, Cave Hill, Barbados, **2** The Warren Alpert Medical School of Brown University, Providence, Rhode Island, United States of America, **3** Faculty of Science and Technology, University of the West Indies, Cave Hill, Barbados, **4** Ministry of Education, Technological and Vocational Training Barbados, Bridgetown, Barbados, **5** The St. Michael School Barbados, Bridgetown, Barbados

* pamela.gaskin@cavehill.uwi.edu

## Abstract

### Background

Wide participation in electronic surveys and reliable reporting of anthropometry can serve to reduce costs associated with monitoring of obesity among adolescents where resources are limited. We conducted a single school pilot study among Caribbean adolescents to assess use of electronic surveys and whether face to face encouragement could promote enrollment. In addition, we assessed students' ability to reliably report simple anthropometry.

### Methods

Students were provided with access to an electronic survey on anthropometry and food preferences regarding school-based food offerings. Responses to survey questions were presented as percentages. A sample of students also had their heights and weights measured after reporting these measures from memory. Intra-class correlation coefficients were used to assess reliability among measurers and Bland-Altman plots, consistency between student reported and recorded anthropometric measures and Support Vector Machine to assess robustness of anthropometry prediction models.

### Results

Response rate to the electronic survey was low (9%). Students were able to interpret questions; open-ended options were inappropriately used 13% of the time. Post survey qualitative responses indicated displeasure with use of school-associated e-mail addresses. Concerns with confidentiality were expressed as well as preference for completion of surveys during school time. Students reliably reported anthropometry most measures fell within the 95% CI of Bland-Altman plots. SVM classified with a prediction accuracy of 95%. Estimates of overweight from recorded and reported measures were similar.

**Data Availability Statement:** All relevant data are within the manuscript and its Supporting information files.

**Funding:** The author(s) received no specific funding for this work.

**Competing interests:** The authors have declared that no competing interests exist.

## Conclusions

Adolescents are able to report simple anthropometry, and this can be used to help with monitoring of growth and overweight. Although they are capable of competently completing electronic surveys, school-based email is an ineffective contact tool. In-person school-based contact and administration of surveys are preferred. Adolescents can reliably report simple anthropometry that can be utilized for estimation of overweight/obesity prevalence. This method can be widely applied.

## Background

Barbados is a resource-scarce country, yet like many developing nations it has problems emanating from a high burden of chronic disease [1, 2]. It is hoped that interventions during youth can help to ameliorate or to prevent the onset of some of these diseases in adulthood [3]. A major challenge is the ability to capture data that would help to characterize groups of individuals who should be specially targeted and would help to assess responses to intervention. Self-reported data are used among adults for such purposes [4, 5]. However, there is debate about adolescents and children's ability to give reliable reports of physical measures and behaviours [6, 7].

Given that children are accessible in school, especially in countries where school is mandatory well into adolescence, it forms a convenient setting for early data capture on individuals. Such data can serve to monitor basic growth and weight status among children and adolescents.

It is well accepted that children and adolescents living in the 21st century are generally au fait with the use of electronic gadgets [8, 9] and prefer using these types of instruments to traditional pen and paper means of communication [10, 11]. Nevertheless, a major challenge is the fact that many electronic surveys have low response rates [12]. How adequate levels of participation in such studies can be achieved is in question, particularly if we are to use them for national data capture among secondary school students [13]. It is also important to understand adolescent's ability to use surveys in this format. Examination of open-ended responses can serve to indicate whether the options offered as answers on surveys would be comprehensive enough to reflect the preferred responses of the participants.

A country wide study among school going adolescents in Barbados is in the planning stages. Its intended purpose is to develop efficient and easy methods for capture of anthropometric and food-related data intended to guide and monitor interventions among adolescents in Barbados. It is hoped that reported physical measures, socio-demographic information and associated factors captured by electronic means can be effectively utilized.

In the current pilot study, we sought to test whether face to face encouragement could prompt use of email and electronic interfaces for data capture amongst adolescents. We also tested whether the students would correctly interpret demographic and anthropometric questions and whether they would appropriately use preformatted answers to survey questions. In addition, we sought to assess barriers to recruitment as well as students' ability to reliably report their heights and weights.

## Methods

Permission to conduct the study was given by the Institutional Review Board of the University of the West Indies/Ministry of Health Barbados. All applicable institutional and governmental

regulations concerning the ethical use of human volunteers were followed during this research. Informed written consent was obtained from parents and assent from children.

The purpose of the current pilot study was to test the methods for a potential country wide study among secondary school children. The school selected for this pilot study is a secondary school in Barbados that attracts a wide cross-section of students as it is centrally located in the capital of the country. Entrance to the school is open to children from any school zone on the island. Tuition fees are covered by the Government as is the case with approximately 95% of schools in Barbados. The race mix of students in the school reflects of the general population which has 92% Afro Caribbean, 2.7% White, 3.1% Mixed, 1.3% Indian, 0.4% Other [14]. This school offers the standard curriculum set by the government of Barbados that includes academic and creative instruction, in addition to athletics. Teachers across secondary schools have a similar qualification profile. The urban-rural differences in Barbados are unremarkable given the close proximity of services and availability of transport to all. This school was therefore deemed suitable for a pilot study. Students' socio-economic status was not assessed.

The study was executed in three sections the "Electronic Survey", the "Focus Groups" and the "Reliability Study". Recruitment for the study occurred between February 5th– 22nd 2019. An information sheet with an attached parental consent form was provided for all students except those in the first form as they were unlikely to have much information on the canteen having just entered the school and were therefore excluded from the study. All eligible students were sent an email message using their school-based email address. This provided the link to the electronic-based survey. Volunteers from selected 2nd, 3rd and 5th forms (Grades 8, 9 and 11) comprised the convenience sample used for the focus groups. A representative sample of the 2nd to 5th forms (Grades 8–11) was selected using probability proportionate to size sampling (PPS) for the anthropometric reliability study.

Prior to the issuing of the electronic survey a senior member of the research team addressed the student body during full assembly which is a mandatory meeting for all students. Students were encouraged to complete the survey and were introduced to members of the research team who would collect the physical measurements and reported anthropometry. The method for logging on to the survey was explained. Students were encouraged to ask questions about the study for clarification at the conclusion of the presentation.

The questionnaire was pretested by two investigators and a research assistant for formatting, typographical errors, whether the questions were able to be read on the electronic medium in a fashion similar to the paper version and if the data would be entered in the database appropriately. In addition, the questionnaire was assessed for face validity. It was then tested by a group of 8 students to determine whether they could easily follow the instructions. There were no reports of misinterpretation or misunderstanding of questions.

## The electronic survey

We developed a survey which was then transposed into electronic format using the SoGoSurvey® software. It started with reconfirmation of consent followed by demographic items, reported anthropometry and questions about the canteen and its food offerings. These items were used to examine whether the electronic interface worked as expected with items appearing in the intended sequence. Demographic questions including date of birth, sex and form were compared with school records as a means of assessing children's ability to interpret questions. Weights and heights were used to assess whether students would correctly use imperial or metric units. We intentionally selected simple subject matter as the basis of most of the survey items; this we believed would enhance our ability to assess use of the tool as the questions would not be challenging.

The principal of the school provided a list of names, date of birth, sex and form in addition to the school-based e-mail address of each student in the list. A study ID number was assigned to each student so that names could be disconnected for dissemination among the study team. Students used a unique code issued to them to prevent one person answering multiple times. Mail merge was used to send the code and the link to the survey. At the beginning of the survey parents were again required to indicate approval of their child/ward completing the survey by selecting "Yes I agree" or "No I do not agree" to the parental permission question.

We used a proforma to categorize the responses to the survey. It captured information on:

1. *Response rate*

2. *Students' ability to understand the demographic questions by comparison of reported answers with school records*

3. *Appropriate use of the options for different measurement units (weight could be reported as pounds or kilograms and height as centimeters or feet and inches)*

4. *Appropriate use of the preformatted answers, these were provided as drop-down menus*

5. *Appropriate use of open-ended sections of the questionnaire (responses were coded and categorized as to whether they conveyed the same message as a preformatted option or were a novel response)*

6. *Whether the SOGO software collated the data in a manner that was suitable for analysis.*

7. *Open-ended responses to be used to refine questionnaires for the larger study.*

*All questions except for the permission and demographics questions were presented in the format in the example below*:

1. *What meals do you eat from the canteen? **(Please choose all that apply)***

    a. *(Breakfast, Morning snack, Lunch, Evening snack)*

2. *I preferred the foods offered by the pervious operator: (Y/N/No preference)*

*Students were offered the following responses to foods offered by the canteen for example fresh fruit*:

3. ***Fresh fruit***:

    a. *How many times a week do you buy it? (Open ended)*

    b. *I buy **Fresh fruit** because*:

        i. *It is cheap*

        ii. *I like it*

    c. *I DO NOT buy **Fresh fruit** because*:

        i. *It is expensive*

        ii. *I do not like it*

        iii. *It is unhealthy*

        iv. *Allergies/intolerance*

        v. *Religious reasons*

## The focus groups

Three post survey focus groups were conducted over the period February 20th to 25th 2019 to qualitatively explore reasons for or against completion or engagement in the survey and to further examine students' understanding of how to use the electronic interface. We elected to do 3 groups of 10 children each. This was in order to reflect the spectrum of opinions across different age ranges. Any student from the selected 2nd, 3rd, and 5th form (grade 8–11 equivalent) was eligible to be a participant. Roughly equal numbers of boys and girls were selected until the target of 10 students was met for each group. They were required to be currently enrolled students of the pilot school.

The focus groups were conducted using a set of prepared open-ended questions as a guide. They lasted approximately 30 minutes. The guide was developed to a) describe barriers to recruitment among adolescents in the school setting b) elicit reasons for not completing the survey and c) to get students recommendations for ways in which to improve the survey and methods for improving recruitment. All interviews were conducted by the principal investigator (PI) in a private area at the school site. The PI had extensive training and experience in conducting interviews for quantitative studies and some experience with conducting focus groups. The guide for the focus groups is given in the S4 File.

**Focus group analysis.** The PI took detailed notes during the focus groups of students' responses. A coding scheme was developed based on categories of responses and a proforma was filled with the results. Most of the survey questions could be answered with simple 'yes' or 'no' as the intent was to determine the manner in which the actual instrument was used rather than to analyze the content of the answers. Students' responses were therefor coded in a binary fashion. All analyses were carried out by the PI.

## Reliability study

An aim of the study was to determine whether students were able to reliably report their weights and heights and to compare estimates of overweight/obesity with those calculated from measurements taken by trained observers. Students were asked to report their heights and weights on the electronic survey. However, since it was possible that uptake of the electronic survey might be inadequate, we took an a priori decision to also collect reported weights and heights as well as weights and heights recorded by trained observers at the school level. This would ensure that we were able to capture anthropometric data on a representative sample of the school, even if that could not be elicited from the electronic study.

The physical measurements were taken during school visits. All eligible students were assigned a number. At the level of the classroom we did a simple random sample to reach the number of students required by the PPS for that form.

Prior to the start of the study, observers were trained on measurements of weights and heights using standard methods [15]. Students were asked to first report their heights and weights from memory. Anthropometry could be reported in imperial or metric units and were all converted to S.I. units. Students' heights and weights were then measured by trained observers using standard methods. Participants were measured in uniform with shoes and socks removed. Standing height was measured to 0.1 cm using a stadiometer (Charder Co. Ltd., HM200P, USA). Weight was measured to 0.5 lbs using a digital scale (Tanita® UM-081, USA) [15]. Two consecutive measures were performed for each anthropometric measure. The mean was used in data analyses. BMI was calculated and converted to z-scores using the WHO growth reference and AnthroPlus® program [16]. AnthroPlus® was also used to calculate overweight/obesity estimates [16].

**The sample.**   We used the entire population of students on the school roll in 2$^{nd}$ to 6$^{th}$ form for the electronic survey. A convenience sample of 30 students 10 each from 2$^{nd}$, 3$^{rd}$ and 5$^{th}$ form was selected for three focus groups. To assess the reliability of reported anthropometric measures, a complete random representative sample of 259 students (124 boys, 135 girls) in 2$^{nd}$ to 5$^{th}$ form (grade equivalent 7–11) was calculated. This was at a power of 95% and incorporated an assumed non-response rate of 50%. Next, we carried out stratified probability proportionate to size sampling to select the students. Once we had the measurements of the prescribed number of students per class determined by our PPS Sample; we allowed any other students that wanted to participate to do so. This policy saw us having 11 extra students participating. Giving us a complete sample of 270 students in the study.

**Statistical analysis.**   We calculated percentages to complete the proforma for the electronic survey. Focus group responses were coded and summarized. To assess the reliability of our trained observers, we used intraclass correlation coefficients (ICC). For estimation of reliability between reported measures (from participant report) and recoded measures (measured by the trained observers) we used the Bland Altman method [17]. We first investigated the data for outliers by use of box and whisker plots for the weight and height variables. We then used Bland-Altman plots of weight, height and body mass index z-score (BMIz) [16] to investigate differences.

The Bland-Altman plot is created from a series of paired data for the same specimens using two different methods of measurement. The plot analysis is a simple way to evaluate a bias between the mean differences, and to estimate an agreement interval, within which 95% of the differences of the second method, compared to the first one. Data can be analyzed either as plots of unit difference or percentage difference. The method defines the intervals of agreements; acceptable limits must be defined a priori, based on clinical necessity, biological considerations or other goals. These statistical limits are calculated by using the mean and the standard deviation (*s*) of the differences between the two measurements. A graphical approach is used to check the assumptions of normality of differences and other characteristics.

In a further step we then applied a Support Vector Machine (SVM), to classify the weight status of a student; using the student reported weight and height as the training variables [18]. The SVM was used as an approach to examine our hypothesis that student self-reported anthropometry could act as a proxy for measures taken by the trained observers. We used a universal Radial Basis Function (RBF) kernel [19] to mitigate overfitting.

$$K(x, x') = \exp\left( -\frac{\|x - x'\|^2}{2\sigma^2} \right)$$

We optimized the SVM both via the fit of the data to the hyperplane and by penalizing the number of samples inside the margin simultaneously. We achieved this by optimizing the Cost (C); where C defines the weight contributed by samples inside the margin to the overall error. Consequently, C adjusts for how hard or soft our large margin classification should be. With a low C, samples inside the margins are penalized less than with a higher C. On one extreme, where C is 0, samples inside the margins are not penalized. This can have the effect of disabling the large margin classification. With an infinite C, there would be hard margins on the other extreme. We tuned our model via ten-fold cross-validation on a set of models of interest. From these models we created a vector of a sequence of C values from 0.1 to 100 with increments of 0.2. We applied a 70/30% split of (training data set/testing data set) for cross validation of reported compared to recorded values. BMIz [16] was used to assign the class labels of thin, normal, overweight or obese to weight status groups. This is a common method for describing the performance of a classification model using a confusion matrix. This is a simple cross-

tabulation of the observed and predicted classes of the data. Results were therefore presented as a confusion matrix to describe the performance of the classification model on a set of test data (reported measures) for which the true values (recorded measures) are known. R statistical language and the Python programming language was used to carry out calculations.

Finally, we assessed the agreement between estimates of overweight from reported and recorded by comparing 95% CI of BMIz derived from AnthroPlus® calculations [16].

## Results

### The electronic survey

Response rate for the electronic survey was 9.1%. Seventy-one (71) (41 girls, 30 boys) students responded to the survey. Mean age was 16.15 ± 2.52. Seventy-five percent of respondents were from 4th to 6th form (grade 10 to 12 equivalent). Of those responding, 49% completed the survey.

Since the response rate was low for the electronic survey we applied a two-sample Kolmogorov-Smirnov test (KS test) to determine whether the sample that answered the survey had age and sex distribution similar to the entire school population. Results of the test were D = 0.16901, p-value = 0.02626.

Further, since we had a representative sample of the school from the reliability study, we used BMIs calculated on this study to compare with the BMIs reported in the electronic survey. Weight and height measures were not available for the entire school population. Of the 71 students who answered the electronic survey, 65 reported weights and heights, compared to 270 students on the reliability study. We carried out an independent samples t-test to assess difference in mean BMI. There was no significant difference in BMI between the two groups (Mean difference; 95% CI: 1.49; *p* = 0.203).

Table 1 summarizes the assessment of the electronic survey. Two surveys had parental refusal, however this question appeared to be misunderstood by one parent as all other questions were answered. Notably, this participant indicated strong negative feeling about the canteen.

With respect to the open-ended questions, 14% (44) were answers that were already provided in the drop-down menu. Forty seven percent (47%) (157) of the responses where the open-ended section was used could be classified as "I don't want to". The reasons that the remaining 39% (129) chose the open option were: I brought it from home (14%) (47); there was no minimum frequency offered (11%) (36); novel answers (8%) (25); neither an "unaware" or "don't know" option was offered (6%) (21).

### The focus groups

The focus groups comprised 5girls and 5 boys from the second year, 5 boys and 5 girls from the third year and 6 girls and 4 boys from the 5th year. The 30 students, mostly identified as Afro Caribbean with 2 of mixed race. They ranged in age from 13–17 years.

Focus group responses following administration of both parts of the study suggested the following:

1. Eighty-six percent (86%) of focus group participants were aware that the survey was being conducted. Most got this information from the study researcher during the information session held at full assembly. Older students said they did not listen to the details given by the researcher nor the principal as they often ignored announcements given in this forum. The younger adolescents said that their interest was stimulated at the information session, but it quickly faded from memory. By and large younger children reported not receiving the survey. Many said they did not use e-mail often especially the one given by the school. Some of them said they could not remember their log-on credentials so were unable to

**Table 1. Use of the electronic survey among adolescents.**

| Measurement/Assessment | Methods | Analysis | Findings |
|---|---|---|---|
| Proportion of students who completed the survey | SOGO software automatically recorded the completion status of each questionnaire | Percentage of complete responses was calculated | 49% (35) of questionnaires were complete. Among the incomplete ones 77% (28) were terminated by Question 17 of 43 questions |
| Participants' ability to understand demographic questions | Records obtained from school principal's office were assigned identification numbers (IDs) and used to compare with answers from students | Percentage of responses from students that correctly matched records were calculated | |
| | | | Sex: {M/F} 100% (71) correct responses |
| | | | Date of birth: {dd/mm/yyyy} 99% (70) correct responses |
| | | | Form (Grade): 100% (71) correct responses |
| Use of imperial or metric units | The units for reported weights and heights were examined for credibility. | Weight | 13% (9) of respondents used kg. 82% (58) used lbs |
| | | | 6% (4) gave no response |
| | | | 1% (1) used the options incorrectly |
| | | Height | 23% (16) of respondents used cm |
| | | | 65% (46) used ft and inches |
| | | | 10% (7) gave no response |
| | | | 4% (3) used the options incorrectly |
| *The canteen/food questions* | | | |
| Responses using preformatted options | Frequency of use of drop-down boxes were used to assess appropriate use | Percentage of responses using the drop-down menu | 92% (3858) of questions were answered using the drop-down menu |
| Skip patterns | | | 100% (4189) of skip patterns were appropriate |
| Open ended questions | Frequency of novel answers among open-ended responses | Percentage of responses that were unavailable in the menu | 8% (25) of open-ended responses were novel |
| | Number of answers that were available in the drop-down menu | Percentage of open-ended responses that were inappropriate | 14% (44) of responses were open ended even though a similar option was available in the menu |

attempt the survey. In contrast older students received the e-mail request to the complete the survey and were able to log-on. They reported checking school e-mail often because they are frequently given homework assignments by this means. None of them cited parental objection as a reason not to complete the survey. Among students attempting the survey 50% did not complete it because it took too long. This was also cited as a reason for omission of questions.

2. Survey questions were well understood, as were the use of the dropdown menus. Adolescents doubted that individual responses would be inaccessible to school officials given that the means of contact was their school-based email. They reported that the main reason for low response rate was failure to use school-based emails. They also reported that use of the school assembly to prompt engagement was unsuitable

3. Student suggestions for improved engagement in the survey included maintenance of an electronic medium to administer surveys. Cellular phones were considered an important medium. Most students reported that classroom administration to groups or classes during school time would be desirable. They also felt that if representatives of the study were present during the administration of the survey that they would be more likely to believe that responses would be confidential. Students said they would have responded better had interfaces such as WhatsApp or Instagram been used to alert them instead of e-mail as they often use this type of social media.

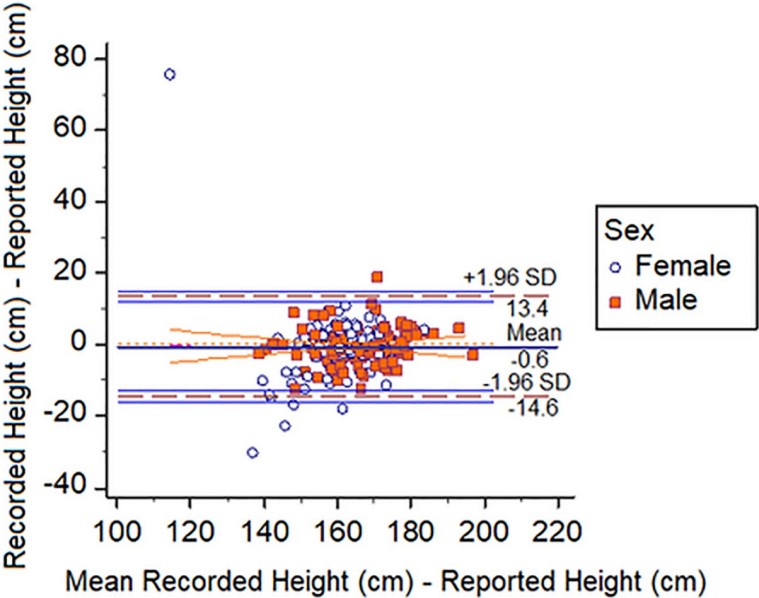

**Fig 1. Bland-Altman plot showing recorded and reported height vs. mean recorded and reported height.** Ninety-nine percent (99%) of all height measures fell within the 95% CI.

## The reliability study

For the reliability study, test retest reliability among raters, was high ICC ≥ 0.97. The response rate was 100% and a representative sample of the school population was easily accessed. The final sample comprised 131 boys and 139 girls, mean age 14.08 ± 1.14 years. Box and whisker plots revealed 7 students on the low end of the height distribution and 5 on the high end. While for weight there were 3 on the high end. The contingency table of outliers for weight status by sex showed that of 12 outliers (6 girls and 6 boys) 9 were on the low end and 3 on the high end.

Bland-Altman plots demonstrated that students reported heights and weights accurately. Figs 1–3 show that 99% of all height measures fell within the 95% CI, this was 93% and 96% for weight and BMIz respectively.

The Bland-Altman plots, by virtue of the almost horizontal lines, demonstrate that there were no statistically significant differences between the pairs of reported and recorded heights and weights and this of course would also be reflected in the BMI comparisons as BMI is calculated using height and weight.

For the SVM, the sample of 270 cases was split into a training dataset (70% of cases) and the remainder of cases in the sample was used by the SVM as a test dataset used to show accuracy of the model's prediction (30% of cases). This allowed the SVM to create the prediction models. The test dataset was used to test how well the SVM predicted weight status (Table 2). The results represent the accuracy of the SVM predictions.

Our SVM model was trained exclusively on student self-reported weight and heights using the recorded data as the gold standard. For this SVM model, we obtained the lowest cross-validation error rate an optimal cost value (C) of 2.2, which gave the performance error of 0.058 and dispersion of 0.046. Using this optimal C of 2.2, we predicted the class labels for weight status on our set of test observations. From the predicted class table, which is the confusion

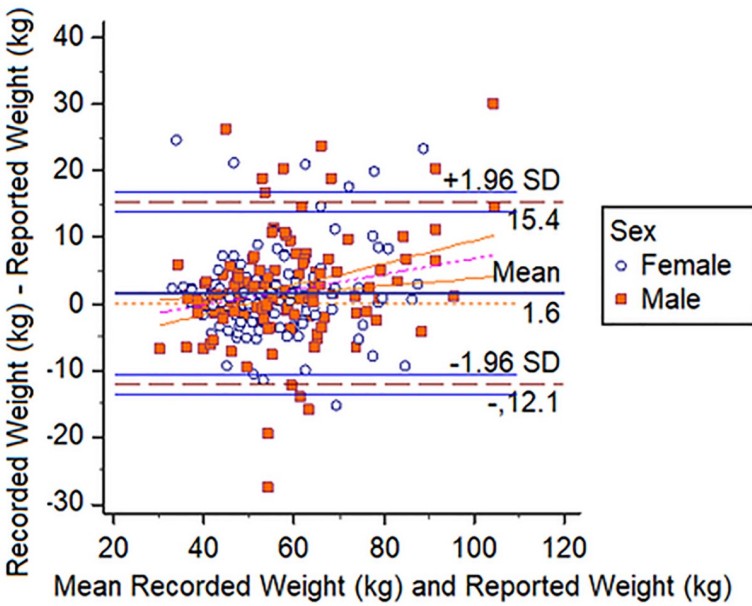

**Fig 2. Bland-Altman plot showing recorded and reported weight vs. mean recorded and reported weight.** Ninety-three percent (93%) of all height measures fell within the 95% CI.

matrix for our test data (Table 2), our SVM classifies students by weight status with a prediction accuracy of 95%.

The table presents the confusion matrix for our test data. The diagonal cells denote cases where the classes are correctly predicted while the off-diagonals illustrate the number of errors for each possible case. The prediction accuracy was 95%.

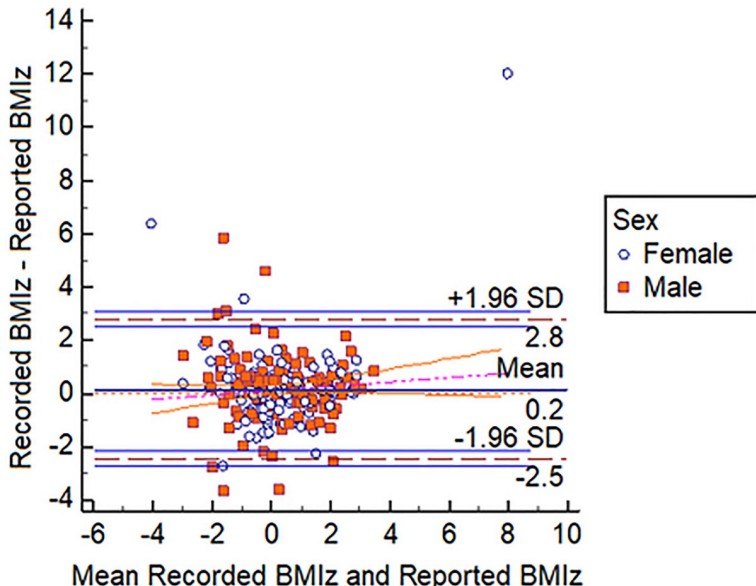

**Fig 3. Bland-Altman plot showing recorded and reported BMIz vs. mean recorded and reported BMIz.** Ninety-six percent (96%) of all height measures fell within the 95% CI.

**Table 2. Support vector machine prediction.**

|  | Normal | Obese | Overweight | Thin |
|---|---|---|---|---|
| Normal | 45 | 0 | 1* | 2* |
| Obese | 0 | 0 | 0 | 0 |
| Overweight | 1* | 2 | 23 | 0 |
| Thin | 0 | 0 | 0 | 8 |

*Misclassified values

**Table 3. Table of distribution of body mass index z score [16] in boys and girls combined with 95% confidence intervals.**

| BMI-for-age (%) (95% CI) | % < -3SD | % < -2SD | % > +1SD | % > +2SD | % > +3SD | Mean | SD |
|---|---|---|---|---|---|---|---|
| Reported measures (n = 270) | 2.6 (0.5%, 4.7%) | 7.4 (4.1%, 10.8%) | 30.5 (24.8%, 36.2%) | 11.2 (7.2%, 15.1%) | 0.7 (0%, 2.0%) | 0.28 | 1.38 |
| Recorded measures (n = 270) | 1.1 (0%, 2.6%) | 2.6 (0.5%, 4.7%) | 29.7 (24.1%, 35.4%) | 10.8 (6.9%, 14.7%) | 2.6 (0.5%, 4.7%) | 0.35 | 1.32 |

Table 3 shows the distribution of BMIz [16]. When the sample was divided into two categories (normal and overweight), 93.7% of students correctly categorized their weight status. The estimates of overweight and obesity for recorded measures were 29.7% (95% CI 24.2–35.2); 10.2 (6.5–14.0) and for reported measures—30 (24.4,35.7): 11 (7.1,14.9).

## Discussion

Parental approval for engagement in the study was high exemplified by the high consent rate among both children completing the electronic survey and those who engaged in the reliability assessment. Findings from our qualitative analyses suggest that administration of electronic surveys was preferred to paper versions and that administration as a classroom activity would significantly raise levels of participation. These factors would address the response rate to the electronic survey a feature which presents a challenge on many surveys using mass email for distribution [20]. Further, our focus group findings suggested that a large group information session outlining the study to students had limited ability to prompt completion of the survey. Notably, students remembered the presentation and younger students reported listening to the contents. In addition, students appeared to have understood what was required as they had no queries about the procedure when instructions were given by the researcher. At the time of the presentation, there was only one question pertaining to the confidentiality of information gathered from students. The issue of confidentiality was also raised in focus groups.

Table 1 shows that students understood how to answer questions using the menus provided and used the "other" option appropriately. Children's ability to answer demographic questions was demonstrated by comparison of records with the official school records. This showed that the method of self-report can be useful. A revised instrument should offer more drop-down menu options which covered items such as the child having brought lunch from home. In addition, it would have been preferable to give a "minimum or less" option for questions that asked frequency of consumption or activity to reduce the use of the open-ended option. Since most students terminated the survey after answering 49% of questions this suggests that the questionnaire was too long.

Given that the D value of the KS statistic was close to 0, it can be inferred that the distribution of age and sex among the sample that answered the electronic survey was not different from the school population. In addition, the children from a representative sample of the school reported similar BMIs to those who reported BMI on the electronic survey. One would

reasonably expect the distribution of BMI on the representative sample to represent that of the school population. This lack of difference in mean BMI supports our conclusion that students who did not complete the electronic survey would be able to use the electronic format to report height and weight. This along with the broad range of students who answered the electronic survey suggests that the survey findings on use of the electronic interface were likely to reflect the general abilities of the student body.

Unlike younger students, older students reported having received the e-mail message requesting participation in the survey. This was reflected by the fact that the mean age for the students responding to the electronic survey was higher than for the other sections of the study. This appeared to the directly related to the fact that school-based e-mail was used in older age groups as a regular method for distribution of homework and other assignments.

Participants said that use of alternate social media platforms such as "WhatsApp" or "Instagram" would have improved their tendency to complete the electronic survey this is supported by finding from other settings [20]. Use of school-based e-mail is likely not optimal and would be unsuitable for collection of sensitive information. Therefore, teacher involvement in organization of small group information presentations and administration of distribution survey should enhance response rates.

Our findings support the notion postulated elsewhere that the mode of recruitment, for example small group sessions, would have greater influence on response rates than does the method of data collection [21]. These may both be enhanced by use of incentives [22].

Students' willingness to engage in the physical survey of body measurements was encouraging. A representative sample of the school was easily gathered. The representative nature of the sample is supported by the fact that the estimate of overweight was in-keeping with expectations [2].

Importantly our Bland Altman plots (Fig 3) demonstrated that 95% of the data points (weight—93%) for the reported anthropometry were within ± 2 SD of the mean difference compared to recorded values. This demonstrated high reliability especially for height. Weight did slightly less well and this would be expected given that weight fluctuates more easily and that body composition tends to have diurnal fluctuations [23]. Bland Altman plots are popularly used to assess accuracy between a gold standard mechanical instrument and an alternative [17]. However, this is not the only application of the technique. In fact, the authors clearly describe its use as a graphical method to compare two clinical measurement techniques [24]. Further, they later described the use of the methodology to assess differences in multiple measurements among human subjects [25]. Hence our findings demonstrate that students in the school we surveyed were capable, of reporting anthropometry accurately. Table 3 clearly shows that estimates of overweight using the two methods were statistically indistinguishable from each other.

We were nevertheless were interested in working through whether reported measures would have application in other schools. We therefore used SVM, a machine learning classification approach, to assess likelihood of consistent high reliability of reported measures. The Bland Altman showed that the students' reporting overall was accurate 97% of the time. Results from our machine learning algorithm therefore enhance the credibility of our findings and indicate that the method would be transferable to other groups with a 95% accuracy.

Our findings demonstrate the potential for use of reported measures collected in electronic format among adolescents. The ability to estimate the weight status of adolescents by relatively cheap swift means, without use of trained observers and long measurement times, has far reaching implications for establishing inexpensive monitoring of changes in pediatric obesity. This, especially as they pertain to school- based obesity interventions in situations where financial and human resources for mounting of large-scale surveys is limited. These methods can be

enhanced by use of: loaner sets of equipment such as scales, stadiometers, wearable activity monitors and simple food intake instruments that would also allow assessments of levels of physical activity and quality of food intake. The high reliability of the overweight estimates and the indications of appropriate ability to use electronic surveys among adolescents under supervised conditions are important as they suggest that simple instruments can be crafted that are easily used by students and can fill a much needed monitoring gap of the distribution of risk factors for obesity and other chronic cardiovascular disease.

## Strengths and limitations

Our use of trained observers to complete the physical measurements added to the integrity of the study. The high a priori intra-class correlation coefficients for height and weight support this.

We were limited to use of a single school to test our hypotheses. However, the school chosen attracts a wide cross-section of students as it is centrally situated. Entrance to the school is open to children from any school zone on the island. Also, socio-economic status has less of an impact on this school as it is falls under the same umbrella of the approximately 95% of secondary schools where tuition fees are covered by the government. In addition, Barbados has a large middle class and a mandatory school leaving age of 16 years that is well enforced. These factors coupled with high literacy rates [26] and universal access to exposure to electronic based learning suggest that children's ability to use electronic media and to answer simple questions on anthropometry and other measures is likely to be relatively evenly distributed among adolescents throughout the island, even among children who are of lower SES. The results of our SVM strongly support these assertions.

This was a pilot so that the submission of incomplete questionnaires was not negative but rather can help us to determine and acceptable questionnaire length. In addition, findings from the focus groups, as well as the high participation in the physical measurements study, indicates that parental consent is not a likely barrier to engagement. Some questions would have been optimized by use of more detailed options in drop-down lists and ability to give a "don't know" response. This was ameliorated by provision of open-ended responses.

Our data set was relatively small this was nevertheless mitigated by optimization of the model and the radial basis function kernel we used in the SVM both via the fit of the data to the hyperplane and by penalizing the number of samples inside the margin simultaneously which works well even with smaller number of data points. In addition, our data suggests that even where there is some variability on the actual weight and height measures, this has little effect on the categorization of weight status when it is collapsed into two categories (normal/ overweight) as weight status is frequently reported. For example, while an overweight student may report somewhat lower weight than was measured, our models tell us that the student is still likely to place themselves in the correct category of weight status which results in good estimates of overweight for the group.

A major strength of this study is the fact that reliability was established by three methods the Bland Altman plots with 95% of values within the 95% CI; the very similar estimates of overweight with strongly overlapping confidence intervals (Table 3) and ultimately the SVM which demonstrated the robustness of the method.

## Conclusions

Large group information sessions have moderate usefulness in sensitizing students about an intended survey. This would be enhanced by dissemination using popular social media platforms. Adolescents were able to correctly interpret questions and instructions presented in

electronic questionnaire format. Electronic delivery of surveys was a preferred method of delivery but there was a preference for administration in small class-room groupings within school hours. Use of a school-based email addresses is a barrier to student engagement. Adolescents can reliably report simple anthropometry that can be utilized for estimation of overweight prevalence. This method can be widely applied.

## Supporting information

**S1 Questionnaire.**
(DOCX)

**S1 File.**
(DOC)

**S2 File.**
(DOCX)

**S3 File.**
(DOCX)

**S4 File.**
(DOCX)

**S5 File.**
(DOCX)

**S1 Data.**
(XLSX)

**S2 Data.**
(XLSX)

**S3 Data.**
(XLS)

**S4 Data.**
(XLSX)

## Acknowledgments

Class of MDSC3900 (UWI) 2019–2020 Justin Ward.

## Author Contributions

**Conceptualization:** Pamela S. Gaskin.

**Data curation:** Yvette Mayers.

**Formal analysis:** Peter Chami.

**Investigation:** Pamela S. Gaskin.

**Methodology:** Pamela S. Gaskin, Peter Chami.

**Project administration:** Pamela S. Gaskin, Patricia Warner, Yvette Mayers.

**Resources:** Tamara Nancoo, Patrick Barrett.

**Software:** Patrick Barrett.

**Supervision:** Pamela S. Gaskin.

**Visualization:** Patricia Warner, Yvette Mayers.

**Writing – original draft:** Pamela S. Gaskin.

**Writing – review & editing:** Pamela S. Gaskin.

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
