## [Decision Letter · Decision Letter 0]

13 May 2020

PONE-D-19-32927

Electronic Based Reported Anthropometry - A Useful Tool for Interim Monitoring of Obesity Prevalence in Developing States

PLOS ONE

Dear Dr. Gaskin,

Thank you for submitting your manuscript to PLOS ONE. After careful consideration, we feel that it has merit but does not fully meet PLOS ONE’s publication criteria as it currently stands. Therefore, we invite you to submit a revised version of the manuscript that addresses the points raised during the review process.

We would appreciate receiving your revised manuscript by Jun 26 2020 11:59PM. To enhance the reproducibility of your results, we recommend that if applicable you deposit your laboratory protocols in protocols.io, where a protocol can be assigned its own identifier (DOI) such that it can be cited independently in the future. For instructions see: http://journals.plos.org/plosone/s/submission-guidelines#loc-laboratory-protocols

We look forward to receiving your revised manuscript.

Kind regards,

Jianhong Zhou

Associate Editor

PLOS ONE

2. Please provide additional details regarding participant consent. In the ethics statement in the Methods and online submission information, please ensure that you have specified (1) whether consent was informed and (2) what type you obtained (for instance, written or verbal). If your study included minors, state whether you obtained consent from parents or guardians. Moreover, please clarify your statement at line 167 ("Consents were not

received for one class in the 2nd  form group which may be an administrative error of collection"), and specify whether those data were excluded from analysis.

3. Please include additional information regarding the survey or questionnaire used in the study and ensure that you have provided sufficient details that others could replicate the analyses. For instance, if you developed a questionnaire as part of this study and it is not under a copyright more restrictive than CC-BY, please include a copy, in both the original language and English, as Supporting Information. Moreover, please include more details on how the questionnaire was pre-tested, and whether it was validated.

4. In your Methods section, please provide additional information about the participant recruitment method and the demographic details of your participants. Please ensure you have provided sufficient details to replicate the analyses such as: a) the recruitment date range (month and year), b) a description of any inclusion/exclusion criteria that were applied to participant recruitment, c) a table of relevant demographic details, d) a statement as to whether your sample can be considered representative of a larger population, e) a description of how participants were recruited, and f) descriptions of where participants were recruited and where the research took place.

5. When reporting the results of qualitative research, we suggest consulting the COREQ guidelines: http://intqhc.oxfordjournals.org/content/19/6/349. In this case, please consider including more information on the number, training and characteristics of interviewers and participants in the focus group study; and how data were coded and analysed.

6. Our internal editors have looked over your manuscript and determined that it is within the scope of our Determinants, Consequences and Management of Obesity  Call for Papers. This collection of papers is headed by a team of Guest Editors for PLOS ONE:Rachel Nugent and Pratibha V. Nerurkar. Additional information can be found on our announcement page: https://collections.plos.org/s/obesity-one.

If you would like your manuscript to be considered for this collection, please let us know in your cover letter and we will ensure that your paper is treated as if you were responding to this call. If you would prefer to remove your manuscript from collection consideration, please specify this in the cover letter.

7. PLOS requires an ORCID iD for the corresponding author in Editorial Manager on papers submitted after December 6th, 2016. Please ensure that you have an ORCID iD and that it is validated in Editorial Manager. To do this, go to ‘Update my Information’ (in the upper left-hand corner of the main menu), and click on the Fetch/Validate link next to the ORCID field. This will take you to the ORCID site and allow you to create a new iD or authenticate a pre-existing iD in Editorial Manager. Please see the following video for instructions on linking an ORCID iD to your Editorial Manager account: https://www.youtube.com/watch?v=_xcclfuvtxQ

Reviewers' comments:

Reviewer's Responses to Questions

**Comments to the Author**

1. Is the manuscript technically sound, and do the data support the conclusions?

Reviewer #1: No

Reviewer #2: Partly

2. Has the statistical analysis been performed appropriately and rigorously? 

Reviewer #1: I Don't Know

Reviewer #2: Yes

3. Have the authors made all data underlying the findings in their manuscript fully available?

Reviewer #1: No

Reviewer #2: No

4. Is the manuscript presented in an intelligible fashion and written in standard English?

Reviewer #1: Yes

Reviewer #2: No

5. Review Comments to the Author

Reviewer #1: This manuscript focused on assessment of the use of electronic surveys, comparison of consistency between reported and recorded anthropometric indices, and robustness of anthropometry prediction models. The topic is interesting due to the comparison between reported and recorded anthropometric indices in adolescents and the issue about the ability for adolescents to give reliable reports. However, I found several important and critical issues, and the results of the experiment are not enough to draw conclusions.

Major:

1. For anthropometric measures, basic demographic characteristic information was need.

2. The present study was focused to test whether the students would correctly interpret demographic and anthropometric questions. But I can’t find the contents on interpretation of demographic questions.

3. In comparison of students’ heights and weights from memory with real heights and weights by trained observers, more explanation about Bland Altman method in Statistical analysis subsection and the results of Bland-Altman plots in Results section should be added. Authors present only “…99% of all height measures…” and captions of three figures in Results section. Readers need more accurate and detailed explanation.

4. I confuse the sample size. In The sample subsection, authors mentioned that “A representative sample of 259 students was selected using probability proportionate to size to assess the reliability of reported anthropometric measures”. What are the results of using this sample size? Tables 2 and 3?

5. In results of SVM experiment (in Table 2), how many samples were used in this experiment? 259 samples? Is this the results of ten-fold cross validation of 259 samples? These contents are very confusing.

6. What is the exact information on Table 2? (confusion metric for multi-classes or number of correct predict samples?). More detailed footnotes were needed.

7. The information of detailed measurement of height and weight is very insufficient. The authors should add the information or configuration of recorded measure by trained physician (the situation of wearing clothes or shoes, equipment, etc).

8. It is recommended to present the contents of the sample section procedure as flow chart.

9. Authors mentioned that “ Test retest reliability among raters, was high ICC ≥0.97.” We can’t find this information on figures or tables.

Minor:

1. Why did you present the responses to the survey questions as percentage? It is necessary to write the exact values in addition to percentage.

2. The contents on captured information (such as Response rate, Students’ ability to understand…, and Fresh fruit) in Electronic survey subsection should be added to the supplementary information.

Reviewer #2: Review of PONE-D-19-32927 Also attached.

Summary: An electronic survey administered in school was designed to obtain health survey information. The survey responses were validated against school records. A SVM model was used to use survey data to predict obesity status.

Review:

General comments: The results are not as good as the authors claim. There are high levels of misreports of body weight and height. Even 5 kg is a high level of error (~10 lbs). Moreover, not listed as a limitation, the school records may not be direct measurements. Even if they came from a doctor’s office, sometimes these are self-reported and not directly measured.

The weights and height were not discussed in the description of the survey. The entire survey should be available as supplementary material.

The most promising outcome is the results of the SVM model which had a 95% accuracy of predicting obesity status.

The authors should also include their code and data sharing statements so that their work could be replicated.

There are many sentence fragments and misspelled words. The authors should clean up the manuscript for these errors. The reviewer lists the following:

Needs a comma and revision suggested inclusions are in red:

Given that children are accessible in school, especially in countries where school is mandatory

well into adolescence, schools form a convenient setting for early data capture of individuals.

Not a sentence - revise:

Nevertheless, a major challenge is the fact that

55 many electronic surveys have low response rates,[12] how adequate levels of participation in

56 such studies can be achieved if we are to use them for national data capture among secondary

57 school students is in question.

Line 61 Hyphenate school-going

Not a sentence:

These comprised questions on the menu of the school

77 canteen and demographics. This

Not a sentence. What does and form mean?

Demographic questions including date of birth, sex and form

6

were compared with school records as 83 a means of assessing children’s ability to interpret

84 questions. Forms 1-6 are equivalent to grades 6-12.

This was approved by an ethical board, but seems coercive. Students will not be likely to not consent if a principal is distributing the survey.

Consent to engage in the study was distributed and collected by the principal who oversaw

154 access to students.

This is the first time we hear that weights are being reported in the survey.

Since it was possible that uptake of the electronic survey might be inadequate, an aim of the

158 study was to determine whether students were able to reliably report their weights and heights

159 and to compare estimates of overweight/obesity with those calculated from measurements taken

160 by trained observers.

Python is misspelled

reported compared to recorded values. BMIz[14] was used to assign the class labels of thin, normal, overweight or obese to weight status groups. R statistical language and the Pyhton programming language was used to carry out calculations.

Enlarge means to make larger. Just state Younger children reported that they did not receive the survey.

By enlarge younger children reported not receiving the survey. Many said they did not

242 use e-mail often especially the one given by the school. Some of them said they could not

243 remember their log-on credentials so were unable to attempt the survey. In contrast older

244 students received the e-mail request to the complete the survey and were able to log-on. They

245 reported checking school e-mail often because they are often given homework assignments by

246 this means. Among students attempting the survey 51%% did not complete it because it took too

247 long. This was also cited as a reason for omission of questions.

Box and whisker plots revealed 7 students on the low end of the height distribution and 5 on the

265 high end. While for weight there were 3 on the high end. The contingency table of outliers for

266 weight status by sex showed that of 12 outliers (6 girls and 6 boys) 9 were on the low end and 3

267 on the high end.

6. PLOS authors have the option to publish the peer review history of their article (what does this mean?). If published, this will include your full peer review and any attached files.

Reviewer #1: No

Reviewer #2: Yes: Diana Thomas

---

## [Author Response · Author response to Decision Letter 0]

25 Jun 2020

We thank the reviewers for their cogent comments to our submission “Electronic-based reported anthropometry - A useful tool for interim monitoring of obesity prevalence in developing states”. We have attempted to revise and improve the manuscript using this guidance. Edits are highlighted using Track Changes in Microsoft Word directly in the text. Below we have responded to the comments of the associate editor and the reviewers in the order in which they were made. Your comments are in Italics and our responses in normal type.

ASSOCIATE EDITOR COMMENTS TO AUTHOR 

1. “Please ensure that your manuscript meets PLOS ONE's style requirements, including those for file naming. The PLOS ONE style templates can be found at http://www.plosone.org/attachments/PLOSOne_formatting_sample_main_body.pdf and http://www.plosone.org/attachments/PLOSOne_formatting_sample_title_authors_affiliations.pdf”

We have amended our manuscript and referencing style by the guidelines shown in the PLOS ONE formatting template.

2. Please provide additional details regarding participant consent. In the ethics statement in the Methods and online submission information, please ensure that you have specified (1) whether consent was informed and (2) what type you obtained (for instance, written or verbal). If your study included minors, state whether you obtained consent from parents or guardians. Written and assent from children Moreover, please clarify your statement at line 167 ("Consents were not

received for one class in the 2nd form group which may be an administrative error of collection"), and specify whether those data were excluded from analysis. 

We apologize for the miscommunication. The principal investigator of the study gave all consent forms to the principal of the school. She had the forms disseminated to student so that they could be disseminated to parents. However, the principal was not actively involved in distribution or collection of consent forms. Only children who had written consent were included in the study. This is clarified at Line 76.

The comment about second form refers to the fact that the 15 consent forms were not returned. The children were therefore not included. We have removed this statement so as not to confuse the reader given that only children with consent were included in the study. All students involved in the study gave assent.

3. Please include additional information regarding the survey or questionnaire used in the study and ensure that you have provided sufficient details that others could replicate the analyses. For instance, if you developed a questionnaire as part of this study and it is not under a copyright more restrictive than CC-BY, please include a copy, in both the original language and English, as Supporting Information. Moreover, please include more details on how the questionnaire was pre-tested, and whether it was validated.

We thank the reviewer for highlighting this omission. We have included greater detail on the way in which we executed the study (Line 95). At Line 113 we have included information on pretesting of the questionnaire and augmented the section on the electronic survey. The questionnaire is included in the Supporting Information.

4. In your Methods section, please provide additional information about the participant recruitment method and the demographic details of your participants. Please ensure you have provided sufficient details to replicate the analyses such as: a) the recruitment date range (month and year), b) a description of any inclusion/exclusion criteria that were applied to participant recruitment, c) a table of relevant demographic details, d) a statement as to whether your sample can be considered representative of a larger population, e) a description of how participants were recruited, and f) descriptions of where participants were recruited and where the research took place.

At line 95 we have included a detailed section on the method of recruitment for all three portions of the study which should allow for repetition of the procedures and have augmented the section on the sample at line 230. We have reorganized the Methods section for better flow.

With regard to inclusion and exclusion criteria, all students attending the school were eligible to be participants in the study except those in the first form (line 95). For the electronic portion of the study, all eligible students (all students except those in first form) were sent an invitation to participate in the survey using their school-based email address. Students are assigned a school-based email address on entry to the school. 

The sample for the reliability study was representative of the distribution of the students by age group and sex. Selection entailed the use of a list of all children present in the school, their form and sex. The method of probability proportionate to size was employed to determine sample size (line 233).

5. When reporting the results of qualitative research, we suggest consulting the COREQ guidelines: http://intqhc.oxfordjournals.org/content/19/6/349. In this case, please consider including more information on the number, training and characteristics of interviewers and participants in the focus group study; and how data were coded and analysed.

We augmented the methods section of the focus group to include information on the interviewer, line 190, participants, line 179, and the analysis, line 195. The guide for the focus groups is given in the Supporting Information. 

6. Our internal editors have looked over your manuscript and determined that it is within the scope of our Determinants, Consequences and Management of Obesity Call for Papers. This collection of papers is headed by a team of Guest Editors for PLOS ONE:Rachel Nugent and Pratibha V. Nerurkar. Additional information can be found on our announcement page: https://collections.plos.org/s/obesity-one.

If you would like your manuscript to be considered for this collection, please let us know in your cover letter and we will ensure that your paper is treated as if you were responding to this call. If you would prefer to remove your manuscript from collection consideration, please specify this in the cover letter.

We would be very happy to have our manuscript included as part of the call for papers on “Determinants, Consequences and Management of Obesity” should you choose to accept it for publication and have indicated such in the cover letter.

7. PLOS requires an ORCID iD for the corresponding author in Editorial Manager on papers submitted after December 6th, 2016. Please ensure that you have an ORCID iD and that it is validated in Editorial Manager. To do this, go to ‘Update my Information’ (in the upper left-hand corner of the main menu), and click on the Fetch/Validate link next to the ORCID field. This will take you to the ORCID site and allow you to create a new iD or authenticate a pre-existing iD in Editorial Manager. Please see the following video for instruction s on linking an ORCID iD to your Editorial Manager account: https://www.youtube.com/watch?v=_xcclfuvtxQ

I have linked my ORCID ID using the Editorial Manager link.

REVIEWER COMMENTS:

1. Is the manuscript technically sound, and do the data support the conclusions?

Reviewer #1: No

Reviewer #2: Partly

2. Has the statistical analysis been performed appropriately and rigorously?

Reviewer #1: I Don't Know

Reviewer #2: Yes

3. Have the authors made all data underlying the findings in their manuscript fully available?

Reviewer #1: No

Reviewer #2: No

4. Is the manuscript presented in an intelligible fashion and written in standard English?

Reviewer #1: Yes

Reviewer #2: No

We have reviewed the manuscript and have revised some sections to improve the flow of the language.

5. Review Comments to the Author

Reviewer #1: This manuscript focused on assessment of the use of electronic surveys, comparison of consistency between reported and recorded anthropometric indices, and robustness of anthropometry prediction models. The topic is interesting due to the comparison between reported and recorded anthropometric indices in adolescents and the issue about the ability for adolescents to give reliable reports. However, I found several important and critical issues, and the results of the experiment are not enough to draw conclusions.

While I respectfully acknowledge that larger sample size is optimal, the close agreement seen in the results of the Bland-Altmann plot as well as the accuracy of the SVM, support our argument for the reliability of the results. In addition, a primary focus of the study was to determine whether overweight estimates could be reliably calculated from report. Our data suggests that even where there is some variability on the actual weight and height measures, this has little effect on the categorization of weight status. Overweight is often expressed in a binary manner (normal or overweight) in this instance the categories thin and normal are collapsed as are overweight and obese. This meant that 93.7% of the students placed themselves in the correct weight status category. We have inserted the percentage of correctly classified students at line 386 in the results section. We have included some discussion of this in the limitations section (line 508).

Major:

1. For anthropometric measures, basic demographic characteristic information was need.

We have included demographic information for the electronic survey at Line 289, for the focus group at line 306 and for the reliability study at line 344.

2. The present study was focused to test whether the students would correctly interpret demographic and anthropometric questions. But I can’t find the contents on interpretation of demographic questions.

Table 1 containing the demographic data has been edited to correctly show the information. The raw data are given in the supplemental file, “SoGoSurvey_The St. Michael school”.

3. In comparison of students’ heights and weights from memory with real heights and weights by trained observers, more explanation about Bland Altman method in Statistical analysis subsection and the results of Bland-Altman plots in Results section should be added. Authors present only “…99% of all height measures…” and captions of three figures in Results section. Readers need more accurate and detailed explanation.

At line 259 we inserted a subsection which gives an explanation of the use of the Bland-Altman plot. We have included a legend on each of the 3 plots.

4. I confuse the sample size. In The sample subsection, authors mentioned that “A representative sample of 259 students was selected using probability proportionate to size to assess the reliability of reported anthropometric measures”. What are the results of using this sample size? Tables 2 and 3?

The representative sample of 259 participants was selected for the reliability study. Figures 1, 2 and 3 as well as Tables 2 and 3 present the results for this section.

5. In results of SVM experiment (in Table 2), how many samples were used in this experiment? 259 samples? Is this the results of ten-fold cross validation of 259 samples? These contents are very confusing.

At Line 364 For the SVM, the sample of 259 cases was split into a training dataset (70% of cases) and the remainder of cases in the sample was used by the SVM as a test dataset used to show accuracy of the model’s prediction (30% of cases). This allowed the SVM to create the prediction models. The test dataset was used to test how well the SVM predicted weight status (Table 2). The results represent the accuracy of the SVM predictions. 

6. What is the exact information on Table 2? (confusion metric for multi-classes or number of correct predict samples?). More detailed footnotes were needed.

We have included a legend in Table 2 and have given a greater explanation in the Statistical Analysis section at line 275

7. The information of detailed measurement of height and weight is very insufficient. The authors should add the information or configuration of recorded measure by trained physician (the situation of wearing clothes or shoes, equipment, etc). 

We apologize for this omission. A section on the measurement procedures is inserted at line 207. Measures were recorded by trained observers.

8. It is recommended to present the contents of the sample section procedure as flow chart.

We inserted sections at Line 95 and at Line 229 which explain the sample selection more clearly.

9. Authors mentioned that “ Test retest reliability among raters, was high ICC ≥0.97.” We can’t find this information on figures or tables.

We have included a supplementary file named “A priori reliabilities” showing inter-rater reliability measurements.

Minor:

1. Why did you present the responses to the survey questions as percentage? It is necessary to write the exact values in addition to percentage.

We have inserted the actual numbers in Table 1. 

2. The contents on captured information (such as Response rate, Students’ ability to understand…, and Fresh fruit) in Electronic survey subsection should be added to the supplementary information.

The response rate for the electronic survey was calculated as a percentage of the total number of responses of the email requests made. For the reliability study, all students selected agreed to engage in the study. We have included the proforma for the electronic study in the file “St. Michael's Canteen Study Electronic Survey Proforma”.

Reviewer #2: Review of PONE-D-19-32927 Also attached.

Summary: An electronic survey administered in school was designed to obtain health survey information. The survey responses were validated against school records. A SVM model was used to use survey data to predict obesity status.

Review:

General comments: The results are not as good as the authors claim. There are high levels of misreports of body weight and height. Even 5 kg is a high level of error (~10 lbs). Moreover, not listed as a limitation, the school records may not be direct measurements. Even if they came from a doctor’s office, sometimes these are self-reported and not directly measured. We did the direct measurements

We have reorganized the manuscript so that the sequence is clearer to the reader. There were three sections to the study: an electronic survey, a focus group and a reliability study. School records were only used as a means of assessing children’s ability to interpret questions on the electronic survey (line 125, Table 1). These were not used to assess reliability of self-report, this was done in the reliability study (line 203). 

While I respectfully acknowledge that larger sample size is optimal, the close agreement seen in the results of the Bland-Altmann plot as well as the accuracy of the SVM, support our argument for the reliability of the results. In addition, a primary focus of the study was to determine whether overweight estimates could be reliably calculated from report. Our data suggests that even where there is some variability on the actual weight and height measures, this has little effect on the categorization of weight status. Overweight is often expressed in a binary manner (normal or overweight) in this instance the categories thin and normal are collapsed as are overweight and obese. This meant that 93.7% of the students placed themselves in the correct weight status category. We have inserted the percentage of correctly classified students at line 385 in the results section. We have included some discussion of this in the limitations section (line 497).

The weights and height were not discussed in the description of the survey. The entire survey should be available as supplementary material.

The weights and heights on the electronic survey were not compared with records. They were used to assess whether students would correctly use imperial or metric units. For the reliability study, all reported measures were taken directly from students and all recorded measures conducted by the field team. We have included the raw data file “Reliability study data” in the Supporting Information.

The most promising outcome is the results of the SVM model which had a 95% accuracy of predicting obesity status.

The authors should also include their code and data sharing statements so that their work could be replicated.

We have included our raw data, “R-Code”, in the Supporting Information.

There are many sentence fragments and misspelled words. The authors should clean up the manuscript for these errors. The reviewer lists the following:

Needs a comma and revision suggested inclusions are in red:

Given that children are accessible in school, especially in countries where school is mandatory

well into adolescence, schools form a convenient setting for early data capture of individuals.

Not a sentence - revise:

Nevertheless, a major challenge is the fact that

55 many electronic surveys have low response rates,[12] how adequate levels of participation in

56 such studies can be achieved if we are to use them for national data capture among secondary

57 school students is in question.

Line 61 Hyphenate school-going

Not a sentence:

These comprised questions on the menu of the school

77 canteen and demographics. This

Not a sentence. What does and form mean?

Demographic questions including date of birth, sex and form

6

were compared with school records as 83 a means of assessing children’s ability to interpret

84 questions. Forms 1-6 are equivalent to grades 6-12.

We edited the text for grammatical and typographical errors.

This was approved by an ethical board, but seems coercive. Students will not be likely to not consent if a principal is distributing the survey. Consent to engage in the study was distributed and collected by the principal who oversaw access to students.

We apologize for our expression, with regard to the principal’s permission. She in no way engaged in any coercive action. Parental permission had already been given for students to engage in the study. The principal simply afforded the investigator access to the school compound and students so that the students engaging in the focus group could be gathered for the activity. We have revised the manuscript to avoid misinterpretation. See Line 95.

This is the first time we hear that weights are being reported in the survey.

Since it was possible that uptake of the electronic survey might be inadequate, an aim of the

158 study was to determine whether students were able to reliably report their weights and heights

159 and to compare estimates of overweight/obesity with those calculated from measurements taken

160 by trained observers.

Thank you for pointing out this omission. Weights and heights on the survey were used to determine whether children would appropriately use imperial or metric units.

Python is misspelled

reported compared to recorded values. BMIz[14] was used to assign the class labels of thin, normal, overweight or obese to weight status groups. R statistical language and the Pyhton programming language was used to carry out calculations.

This was corrected.

Enlarge means to make larger. Just state Younger children reported that they did not receive the survey.

By enlarge younger children reported not receiving the survey. Many said they did not

242 use e-mail often especially the one given by the school. Some of them said they could not

243 remember their log-on credentials so were unable to attempt the survey. In contrast older

244 students received the e-mail request to the complete the survey and were able to log-on. They

245 reported checking school e-mail often because they are often given homework assignments by

246 this means. Among students attempting the survey 51%% did not complete it because it took too

247 long. This was also cited as a reason for omission of questions.

Box and whisker plots revealed 7 students on the low end of the height distribution and 5 on the

265 high end. While for weight there were 3 on the high end. The contingency table of outliers for

266 weight status by sex showed that of 12 outliers (6 girls and 6 boys) 9 were on the low end and 3

267 on the high end.

We edited this.

---

## [Decision Letter · Decision Letter 1]

9 Sep 2020

PONE-D-19-32927R1

Electronic based reported anthropometry - A useful tool for interim monitoring of obesity prevalence in developing states

PLOS ONE

Dear Dr. Gaskin,

Thank you for submitting your manuscript to PLOS ONE. After careful consideration, we feel that it has merit but does not fully meet PLOS ONE’s publication criteria as it currently stands. Therefore, we invite you to submit a revised version of the manuscript that addresses the points raised during the review process.

We look forward to receiving your revised manuscript.

Kind regards,

Mauro Lombardo

Academic Editor

PLOS ONE

Reviewers' comments:

Reviewer's Responses to Questions

**Comments to the Author**

1. If the authors have adequately addressed your comments raised in a previous round of review and you feel that this manuscript is now acceptable for publication, you may indicate that here to bypass the “Comments to the Author” section, enter your conflict of interest statement in the “Confidential to Editor” section, and submit your "Accept" recommendation.

Reviewer #1: All comments have been addressed

Reviewer #2: All comments have been addressed

2. Is the manuscript technically sound, and do the data support the conclusions?

Reviewer #1: Yes

Reviewer #2: Partly

3. Has the statistical analysis been performed appropriately and rigorously? 

Reviewer #1: I Don't Know

Reviewer #2: No

4. Have the authors made all data underlying the findings in their manuscript fully available?

Reviewer #1: (No Response)

Reviewer #2: No

5. Is the manuscript presented in an intelligible fashion and written in standard English?

Reviewer #1: Yes

Reviewer #2: (No Response)

6. Review Comments to the Author

Reviewer #1: This revised manuscript provided answers to important issues. Major comments have been addressed. So, I hope to accept this manuscript.

Reviewer #2: The reviewer appreciates the responses and has a couple questions.

First, the survey is still not included. The survey should be included.

Second, the response rate was very low. What the authors have actually shown is that the sample that responded had reasonable agreement with the measurements. What are the characteristics of the sample that did not respond? If the authors cannot demonstrate that the individuals who did not respond are different at least in base characteristics, they cannot make a strong statement that all adolescents can supply height and weight by survey.

7. PLOS authors have the option to publish the peer review history of their article (what does this mean?). If published, this will include your full peer review and any attached files.

Reviewer #1: No

Reviewer #2: **Yes: **Diana Thomas

---

## [Author Response · Author response to Decision Letter 1]

28 Sep 2020

September 23, 2020,

Academic Editor

PLOS ONE

Dear Dr. Lombardo,

We thank the reviewers for their cogent comments to our submission “Electronic-based reported anthropometry - A useful tool for interim monitoring of obesity prevalence in developing states”. We have attempted to revise and improve the manuscript using this guidance. Edits are highlighted using Track Changes in Microsoft Word directly in the text. Below we have responded to the comments of the editor and the reviewers in the order in which they were made. Your comments are in Italics and our responses in normal type.

ACADEMIC EDITOR COMMENTS TO AUTHOR 

1. “Thank you for submitting your manuscript to PLOS ONE. After careful consideration, we feel that it has merit but does not fully meet PLOS ONE’s publication criteria as it currently stands. Therefore, we invite you to submit a revised version of the manuscript that addresses the points raised during the review process.”

We have made every attempt to comply with the publication criteria of PLOS ONE. As such, we have responded to reviewers and added additional supporting material.

REVIEWER COMMENTS:

1. If the authors have adequately addressed your comments raised in a previous round of review and you feel that this manuscript is now acceptable for publication, you may indicate that here to bypass the “Comments to the Author” section, enter your conflict of interest statement in the “Confidential to Editor” section, and submit your "Accept" recommendation.

Reviewer #1: All comments have been addressed

Reviewer #2: All comments have been addressed

2. Is the manuscript technically sound, and do the data support the conclusions?

Reviewer #1: Yes

Reviewer #2: Partly

We have added the raw data from the electronic survey in the supporting information in a file labelled “The Electronic Survey raw data”. 

3. Has the statistical analysis been performed appropriately and rigorously?

Reviewer #1: I Don't Know

Reviewer #2: No

We have reviewed the statistical analysis and we believe it to be appropriate and rigorous.

4. Have the authors made all data underlying the findings in their manuscript fully available?

The PLOS Data policy requires authors to make all data underlying the findings described in their manuscript fully available without restriction, with rare exception (please refer to the Data Availability Statement in the manuscript PDF file). The data should be provided as part of the manuscript or its supporting information or deposited to a public repository. For example, in addition to summary statistics, the data points behind means, medians and variance measures should be available. If there are restrictions on publicly sharing data—e.g. participant privacy or use of data from a third party—those must be specified.

Reviewer #1: (No response)

Reviewer #2: No

We have added additional supporting information giving the raw data for the electronic survey and the age/sex distribution of all students in the school in files labelled “The Electronic Survey raw data” and “List of all children by form sex and dob” respectively.

5. Is the manuscript presented in an intelligible fashion and written in standard English?

Reviewer #1: Yes

Reviewer #2: (No Response)

6. Review Comments to the Author

Reviewer #1: This revised manuscript provided answers to important issues. Major comments have been addressed. So, I hope to accept this manuscript.

Reviewer #2: The reviewer appreciates the responses and has a couple questions.

First, the survey is still not included. The survey should be included.

Second, the response rate was very low. What the authors have actually shown is that the sample that responded had reasonable agreement with the measurements. What are the characteristics of the sample that did not respond? If the authors cannot demonstrate that the individuals who did not respond are different at least in base characteristics, they cannot make a strong statement that all adolescents can supply height and weight by survey.

1. We apologize for this omission. We have added the raw data from the electronic survey in the file “The Electronic Survey raw data” in the supporting information.

2. While we admit that the response rate to the survey was low, we have no reason to assume that those who did not answer were unable to supply weight and height by electronic survey. We had baseline age and sex for each student and we have included results at line 293 that which demonstrate that the students who answered the electronic survey were from the same distribution as those in the entire school. At line 425 we discussed this. In addition, do recall that on the reliability study, where we had a representative sample of the school, all children were able to give a credible estimate of their weight and height. We have no reason to believe that they would not have been able to write down these same reported weights and heights on an electronic survey. Also, during the focus groups none of the students reported difficulty with understanding how to answer the survey as a reason for not responding to the electronic survey. 

7. PLOS authors have the option to publish the peer review history of their article (what does this mean?). If published, this will include your full peer review and any attached files.

Do you want your identity to be public for this peer review? For information about this choice, including consent withdrawal, please see our Privacy Policy.

Reviewer #1: No

Reviewer #2: Yes: Diana Thomas

Yours faithfully,

Pamela Gaskin PhD 

Lecturer, Essential National Health Research

---

## [Decision Letter · Decision Letter 2]

6 Oct 2020

PONE-D-19-32927R2

Electronic based reported anthropometry - A useful tool for interim monitoring of obesity prevalence in developing states

PLOS ONE

Dear Dr. Gaskin,

Thank you for submitting your manuscript to PLOS ONE. After careful consideration, we feel that it has merit but does not fully meet PLOS ONE’s publication criteria as it currently stands. Therefore, we invite you to submit a revised version of the manuscript that addresses the points raised during the review process.

We look forward to receiving your revised manuscript.

Kind regards,

Mauro Lombardo

Academic Editor

PLOS ONE

Reviewers' comments:

Reviewer's Responses to Questions

**Comments to the Author**

1. If the authors have adequately addressed your comments raised in a previous round of review and you feel that this manuscript is now acceptable for publication, you may indicate that here to bypass the “Comments to the Author” section, enter your conflict of interest statement in the “Confidential to Editor” section, and submit your "Accept" recommendation.

Reviewer #2: All comments have been addressed

2. Is the manuscript technically sound, and do the data support the conclusions?

Reviewer #2: Yes

3. Has the statistical analysis been performed appropriately and rigorously? 

Reviewer #2: Yes

4. Have the authors made all data underlying the findings in their manuscript fully available?

Reviewer #2: Yes

5. Is the manuscript presented in an intelligible fashion and written in standard English?

Reviewer #2: (No Response)

6. Review Comments to the Author

Reviewer #2: It's not satificatory to say that we have no reason to believe there is no difference between those who filled the survey or not. Make a table of the non-survey takers and the survey takers with statistical characteristics. Is BMI different between groups? Even better is to run a t-test.

7. PLOS authors have the option to publish the peer review history of their article (what does this mean?). If published, this will include your full peer review and any attached files.

Reviewer #2: No

---

## [Author Response · Author response to Decision Letter 2]

28 Oct 2020

Dear Dr. Lombardo,

We thank the reviewers for their cogent comments to our submission “Electronic-based reported anthropometry - A useful tool for interim monitoring of obesity prevalence in developing states”. We have attempted to revise and improve the manuscript using this guidance. Edits are highlighted using Track Changes in Microsoft Word directly in the text. Below we have responded to the comments of the editor and the reviewers in the order in which they were made. Your comments are in Italics and our responses in normal type.

Reviewers' comments:

Reviewer's Responses to Questions

Comments to the Author

1. If the authors have adequately addressed your comments raised in a previous round of review and you feel that this manuscript is now acceptable for publication, you may indicate that here to bypass the “Comments to the Author” section, enter your conflict of interest statement in the “Confidential to Editor” section, and submit your "Accept" recommendation.

Reviewer #2: All comments have been addressed

2. Is the manuscript technically sound, and do the data support the conclusions?

Reviewer #2: Yes

3. Has the statistical analysis been performed appropriately and rigorously?

Reviewer #2: Yes

4. Have the authors made all data underlying the findings in their manuscript fully available?

Reviewer #2: Yes

5. Is the manuscript presented in an intelligible fashion and written in standard English?

Reviewer #2: (No Response)

6. Review Comments to the Author

Reviewer #2: It's not satisfactory to say that we have no reason to believe there is no difference between those who filled the survey or not. Make a table of the non-survey takers and the survey takers with statistical characteristics. Is BMI different between groups? Even better is to run a t-test.

We addressed the comparison of BMI differences among those who filled the survey compared to non-survey takers at line 296 in the results section. Given that we did not have weights and heights for the entire school population we used BMIs reported on the reliability study, which had a representative sample of the school, to compare with the BMIs reported in the electronic survey. It is reasonable to expect that the distribution of the BMI on the representative sample would be very similar to that of the school.

Of the 71 students who answered the electronic survey, 65 reported weights and heights, compared to 270 students on the reliability study. We carried out an independent samples t-test to assess whether the mean reported BMI on the electronic survey was significantly different from that of the representative sample. There was no significant difference in BMI between the two groups (Mean difference; 1.49; p = 0.203). 

Do recall that a priori we expected a low response rate so that the electronic survey was never intended to be used as a means of assessing children’s ability to report anthropometry reliably. Rather it was used to explore ability to complete the survey and to help us to understand barriers and solutions to completion of surveys presented in this format.

7. PLOS authors have the option to publish the peer review history of their article (what does this mean?). If published, this will include your full peer review and any attached files.

Do you want your identity to be public for this peer review? For information about this choice, including consent withdrawal, please see our Privacy Policy.

Reviewer #2: No

Yours sincerely

Pamela Gaskin PhD 

Lecturer, Essential National Health Research

---

## [Decision Letter · Decision Letter 3]

18 Nov 2020

Electronic based reported anthropometry - A useful tool for interim monitoring of obesity prevalence in developing states

PONE-D-19-32927R3

Dear Dr. Gaskin,

We’re pleased to inform you that your manuscript has been judged scientifically suitable for publication and will be formally accepted for publication once it meets all outstanding technical requirements.

Kind regards,

Mauro Lombardo

Academic Editor

PLOS ONE

Additional Editor Comments (optional):

Reviewers' comments:

Reviewer's Responses to Questions

**Comments to the Author**

1. If the authors have adequately addressed your comments raised in a previous round of review and you feel that this manuscript is now acceptable for publication, you may indicate that here to bypass the “Comments to the Author” section, enter your conflict of interest statement in the “Confidential to Editor” section, and submit your "Accept" recommendation.

Reviewer #2: All comments have been addressed

2. Is the manuscript technically sound, and do the data support the conclusions?

Reviewer #2: Yes

3. Has the statistical analysis been performed appropriately and rigorously? 

Reviewer #2: Yes

4. Have the authors made all data underlying the findings in their manuscript fully available?

Reviewer #2: Yes

5. Is the manuscript presented in an intelligible fashion and written in standard English?

Reviewer #2: Yes

6. Review Comments to the Author

Reviewer #2: The authors have addressed this reviewer's concerns with patience and grace. Thank you. This form is making me go beyond 100 characters so I filled it with a water sandwich.

7. PLOS authors have the option to publish the peer review history of their article (what does this mean?). If published, this will include your full peer review and any attached files.

Reviewer #2: No

---

## [Editor Report · Acceptance letter]

25 Nov 2020

PONE-D-19-32927R3 

Electronic based reported anthropometry - A useful tool for interim monitoring of obesity prevalence in developing states 

Dear Dr. Gaskin:

I'm pleased to inform you that your manuscript has been deemed suitable for publication in PLOS ONE. Congratulations! Your manuscript is now with our production department. 

Kind regards, 

on behalf of

Dr. Mauro Lombardo 

Academic Editor

PLOS ONE